# Study on Rapid Detection of Pesticide Residues in Shanghaiqing Based on Analyzing Near-Infrared Microscopic Images

**DOI:** 10.3390/s23020983

**Published:** 2023-01-14

**Authors:** Haoran Sun, Liguo Zhang, Lijun Ni, Zijun Zhu, Shaorong Luan, Ping Hu

**Affiliations:** Chemistry & Molecular Engineering College, East China University of Science & Technology, Shanghai 200237, China

**Keywords:** pesticide residues, near-infrared microscopic imaging, rapid detection, computer vision, Shanghaiqing

## Abstract

Aiming at guiding agricultural producers to harvest crops at an appropriate time and ensuring the pesticide residue does not exceed the maximum limit, the present work proposed a method of detecting pesticide residue rapidly by analyzing near-infrared microscopic images of the leaves of Shanghaiqing (Brassica rapa), a type of Chinese cabbage with computer vision technology. After image pre-processing and feature extraction, the pattern recognition methods of K nearest neighbors (KNN), naïve Bayes, support vector machine (SVM), and back propagation artificial neural network (BP-ANN) were applied to assess whether Shanghaiqing is sprayed with pesticides. The SVM method with linear or RBF kernel provides the highest recognition accuracy of 96.96% for the samples sprayed with trichlorfon at a concentration of 1 g/L. The SVM method with RBF kernel has the highest recognition accuracy of 79.16~84.37% for the samples sprayed with cypermethrin at a concentration of 0.1 g/L. The investigation on the SVM classification models built on the samples sprayed with cypermethrin at different concentrations shows that the accuracy of the models increases with the pesticide concentrations. In addition, the relationship between the concentration of the cypermethrin sprayed and the image features was established by multiple regression to estimate the initial pesticide concentration on the Shanghaiqing leaves. A pesticide degradation equation was established on the basis of the first-order kinetic equation. The time for pesticides concentration to decrease to an acceptable level can be calculated on the basis of the degradation equation and the initial pesticide concentration. The present work provides a feasible way to rapidly detect pesticide residue on Shanghaiqing by means of NIR microscopic image technique. The methodology laid out in this research can be used as a reference for the pesticide detection of other types of vegetables.

## 1. Introduction

As an important part of agricultural production, pesticides play an active role in crop growth [1]. With population growth and rising demand for agricultural products, agricultural producers are becoming increasingly more dependent on pesticides, leading to a rapid increase in pesticide use and even cases of overuse [2]. Although pesticide use can increase crop production, it also brings many problems, such as food safety, environmental pollution and so on. Pesticide residues in crops have a serious impact on human health and natural ecosystems [3,4,5]. Since pesticide use is unavoidable and effective, rapid methods of pesticide testing and harvesting crops at a suitable time are essential to food safety.

Common domestic and international pesticide testing is conducted mainly on the basis of chemical or biological principles. Thin layer chromatography (TLC) is a general method for the detection of trace matter [6]. The method is simple in laboratory environment, but the accuracy is poor. It is often used for qualitative detection. Gas chromatography (GC) is most frequently used in pesticide residue detection. It provides precise detection of substances at the molecular level. Therefore, it is often applied to calibrate other methods for detecting pesticide residues. For substances with excessively high boiling points and poor thermal stability, high performance liquid chromatography (HPLC) is a common method. In the field of food analysis, such as vegetables, fruits, edible mushrooms, etc., GC and HPLC can be combined with instruments, such as precolumn extraction, purification, and inductively coupled plasma mass spectrometry. However, both GC and HPLC have shortcomings, such as large solvent consumption, lengthy analysis, high maintenance costs, and the need for professional operators. They are not suitable for rapid detection on crop picking sites or the market. Enzyme inhibition is a well-established technique for rapid detection of pesticides [7] and is widely used due to its low cost and simple detection operation [8]. Although the method is rapid, except for organophosphorus and carbamate pesticides, it is not suitable for detecting other types of pesticides that have no obvious inhibition on the enzyme. In such cases, it is necessary to pretreat the sample or add oxidants to improve the detection sensitivity of the enzyme inhibition method. Biosensors have the advantages of miniaturization, high sensitivity, rapidity, reliability, and ease of use [9]. Various biosensors are being applied to detect pesticide residue rapidly [10]. For example, Ana et al. summarized 57 articles studying nanomaterial-based biosensors for pesticide detection in foods [11]; they found that all biosensors developed in the selected investigations had a limit of detection (LODs), lower than the Codex Alimentarius maximum residue limit, and were efficient in detecting pesticides in food. Ulas et al. developed SMART (specific, measurable, accurate, robust, and time-saving) biosensors [12] which can detect toxic small molecules, such as antibiotics, pesticides, and insecticides, to overcome the shortcomings of traditional HPLC and MS, such as long sample preparation time and cumbersome instruments. Molecularly imprinted electrochemical sensor was successfully applied in the detection of cyromazine residues in agricultural products [13]. However, because biosensor detection requires specific binding to this process, the pesticide species needs to be roughly identified before the detection, and this property reduces the detection efficiency of the methods.

At the beginning of the 21st century, computer vision (CV) technology began to be applied in many fields because it is low cost, easy to operate, and returns rapid results [14]. CV uses computers to analyze images and perform purposeful processing of the information contained in them [15]. Together with pattern recognition methods, the images can be classified into given categories. With the further development of this technology, CV technology has become one of the technologies that run through the agricultural industry chain [16]. For example, Sankar et al. [17] developed a low-cost paper-based biosensor for rapid determination of chlorpyrifos with high accuracy and low cost for in situ detection of chlorpyrifos with CV techniques. Ren et al. [18] used a hyperspectral imaging system to obtain image information from 900 to 1700 nm of spinach samples sprayed with different concentrations of dimethoate emulsion. The discriminant model was established by chi-square test and linear discriminant analysis (LDA), and the prediction accuracy of the pesticide residues of Legoland could reach 99.70%. Jiang et al. [19] used a CV-based image segmentation algorithm to process hyperspectral images of pesticide-sprayed and non-pesticide-sprayed apples. A classification model was developed by using convolutional neural networks, and the discrimination accuracy of the test set reached 99.09%. The pesticide residue detection methods related to CV technology provide nondestructive and environmental friend detection methods of pesticide residues. However, the amount of spectral data of hyperspectral imaging is huge, the computation workload increases exponentially, and the overall detection speed is slow. In addition, the hyperspectral imaging instrument is expensive and bulky, which is not suitable for rapid detection application in crop picking sites or the market. While obtaining images by means of biosensors, it needs to know the kind of pesticides prior to detection. Therefore, applying CV technology to detect pesticide residue of crops rapidly must find a way of obtaining informative images at a low cost and with instruments that can be applied on spot. 

Currently, most light sources used for pesticide residue detection with CV technology are white light [20]. It is well known that different substances have different absorbance under specific light source. Therefore, given any substance, there is a light source most suitable for providing more clear images with good interference resistance. On the basis of such types of images, the effectiveness of CV technology can be improved significantly. 

Near-infrared (NIR) spectra are vibrational spectra that can be used to characterize unknown pesticides [21]. With the development of computer and chemometrics, near-infrared microscopic imaging technology has been more and more widely used in many fields [22]. NIR microscopic imaging technology combines NIR spectroscopy instruments with microscopic imaging collection and CV technology to conduct qualitative and quantitative analysis of samples. However, the molecular structures of the pesticide, the shape properties of the sample, and the wavelength of light source will affect the image properties. In addition to the pretreatment of the images, qualitative and quantitative (pattern recognition) methods also affect the analysis results of NIR microscopic imaging. Therefore, for a specific sample, pesticide, light source, and image pretreating method, the analyzing methods of images should be inspected. 

Crop leaves are composed mainly of cellulose in the leaf pulp and epidermis, and cellulose contains mainly hydroxyl, ether bond, and other groups, which are quite different from the groups in pesticides. For example, the commonly used pesticide cypermethrin contains mainly C-Cl, C=C, -CN, C=O, C-O-C, -C_6_H_5_, and other groups. These groups all have characteristic absorption at different levels in the infrared and visible light region. The characteristic absorption positions and absorption intensities of pesticides with different chemical structures are also different [23]. Therefore, the resolvability of the (microscopic) images of vegetables’ surfaces changes with wavelengths of the light sources. The wavelength will influence the clarity and interference resistance of (microscopic) images of agricultural products sprayed (or not) with pesticides, which will ultimately impact the effectiveness of pattern recognition that is based on crop images. 

Given the above considerations, this work considers pesticide residue detection of sprayed by trichlorfon and cypermethrin as the case study. The appropriate wavelengths are screened to find the wavelength of light source to obtain images that can differentiate Shanghaiqing (Brassica rapa) from those sprayed with the two types of pesticides. A portable visible-near-infrared microscopic imaging device, which was designed to obtain the microscopic images of Shanghaiqing (sprayed and unsprayed with pesticides), was developed in this work. Then, the images were further processed to extract the classification features on which recognition models were built to assess the presence of the pesticides on the Shanghaiqing.

In addition, a model of predicting residual pesticide concentration on Shanghaiqing was established by multiple regression, with cypermethrin concentration as the dependent variables and the image features as independent variables to estimate initial concentration of cypermethrin sprayed on Shanghaiqing. Then, the degradation equation of cypermethrin was built. On the basis of the multiple regression equation and the degradation equation, the time it takes for the cypermethrin concentration to reduce to an acceptable level can be estimated.

## 2. Materials and Methods

### 2.1. Research Roadmap of Using NIR Microscopic Imaging Technique to Detect Residual Pesticides on Vegetables Rapidly

The NIR microscopic imaging technology used to detect residual pesticides on vegetables surfaces in this study contains several steps, as shown in Figure 1. 

The first step of this study is to design a device to acquire microscopic images of samples. The absorbance of the groups in different substances are different at the same wavelength, while the absorbance of the groups of the same substance varies with wavelengths. Therefore, for a given pesticide, selecting a light source with suitable wavelength will be helpful for obtaining high-resolution images. The second step is to screen for the most suitable wavelength of the NIR light source used in the device. Once the wavelength is determined, in the third step, the NIR microscopic images of the leaves sprayed or unsprayed with the pesticides are acquired under the NIR light at the selected wavelength. In the fourth step, the images are preprocessed to remove noise and uninformative signals and segmented to extract image features. The fifth step is divided into two parts. One is to build pattern recognition models with four classification methods on the basis of the image features to assess whether the samples were sprayed with the pesticides. Another one is to establish the relationship between the concentration of the residual pesticide and the extracted image features to estimate the initial pesticide concentration on the leaves of green vegetables. Finally, a degradation equation of the pesticide is established. On the basis of the estimated initial pesticide concentration and the degradation equation, the degradation time can be estimated to guide agricultural producers in harvesting vegetables at an optimized time. 

### 2.2. Design and Installation of Near-Infrared Microscope Imaging System

In order to acquire NIR microscope images of Shanghaiqing, in the present work, an installation of the NIR microscope imaging system is designed, whose mechanical structure is shown in Figure 2.

The main components of the near-infrared microscopy imaging system consist of a CMOS industrial camera 203, a micro-lens 202 with a filter, and LED light arrays 121 in addition to a sample holder 201 and light source cassette 112. The sample holder 201 is used to hold the sample by inserting two glass slides with leaf samples pressed into it. The light source cassette 121 is used to hold the NIR light source and is prepared from the same material as the sample holder by 3D printing. On the side of the cassette are openings connected to the power input system 111 with switches 113. One switch is used for changing the wavelength of the light and another one for turning on the light. The micro-lens 202 is used to broaden the spatial domain of the sample surface, and the lens selected in this work can magnify the images by 800 times. The capture process converts the optical signals into electrical signals and finally into digital signals. A CMOS camera is used in this system for comprehensive consideration on cost. The lens selection should consider the working distance, imaging size, CMOS image element size, and resolution. The CMOS image element size chosen for this system is 2.2 μm × 2.2 μm, the working distance is 90~100 mm, and the camera lens interface is CS type interface.

The power input system 111 used in this system has an input voltage of 100–240 V, an AC frequency of 50 Hz or 60 Hz, an output voltage of 9 V, and a maximum output current of 1 A. It can be connected with conventional with electricity. The complementary light system includes the LED light source as well as the PCB control system. The wavelength range of the LED light source is 430~935 nm, which covers the visible and near infrared light ranges. The microscopic images of Shanghaiqing samples sprayed with 0.1 g/L cypermethrin were taken at 430 nm, 470 nm, 560 nm, 660 nm, and 935 nm wavelengths, respectively. The segmentation results of these images are shown in Figure 3. It indicates that the Figure 3e separates the areas sprayed pesticides droplets from those without pesticide droplets clearly, and it has more obvious image feature of pesticide droplets than the other figures. Therefore, the micro-images taken under the light source at 935 nm are more readily identifiable than those taken at other wavelength light sources. Therefore, infrared light source at 935 nm is selected as the light source of this study.

### 2.3. Sample Set Partition, Pre-Processing, and Segmentation of the Infrared Microscopic Images

#### 2.3.1. Sample Set Partition 

Prior to building a classification models, the samples were divided into training and validation sets. The methods of partitioning samples include random method, KS method [24], and SPXY method [25]. In this work, the KS method was applied to select the training set from the sample set. The KS method selects the two samples having the largest Euclidean distance firstly in to the training set. Then, we calculated the Euclidean distance of each remnant sample to the center of the training set, found the samples with the maximum and minimum distances to the center, and assigned them to the training set until the required number of training samples was reached. The impact of the ratio between the training samples and the validation samples on model accuracy was inspected. 

#### 2.3.2. Pre-Processing of the Images

The initial images often involve many uninformative signals, such as noises and interference coming from environments and so on. Image preprocessing can improve the quality of images, subsequent data detectability, and the accuracy of subsequent image recognition.

In this work, the captured color near-infrared microscopic images were grayed firstly to visually enhance the contrast and highlight the target area. Therefore, the color information of the image was removed and only the grayscale information was used in the subsequent image treatment and recognition. Furthermore, Gaussian filtering was used to remove and suppress the noises, and the images were convolved by Gaussian function. Moreover, the Retinex method was used for enhancing the images to reduce the effect of uneven illumination, enhance the efficiency of subsequent image segmentation, and highlight the region sprayed with pesticides. The method includes three steps: data preparation, calculation of relative light and dark in each band, and data display. Retinex method considers that the image is composed of reflection image component and brightness image component, and the image is not affected by non-uniformity of illumination, i.e., it thinks the color of image is invariant. The advantage of Retinex method is that it can reach a balance among dynamic compression, edge enhancement, and color constancy and, thus, achieves uniform lighting [26]. Then, the images were processed by closed operation to fill the tiny holes in the area of pesticide droplets and amplify its features.

#### 2.3.3. Image Segmentation

The segmentation of the near-infrared microscopic images is to separate the areas sprayed with pesticide droplets from the areas without droplets in the image. Because the leaf veins contain little chlorophyll and have only ducts and sieve tubes, they differ from the fleshy parts of the leaves, and the leaf vein region needs to be removed by using image segmentation. In this study, the Niblack [27] local threshold method together with the connected component threshold method was applied to segment the images. The idea of Niblack algorithm is to use a window with fixed size to slide over the image using each pixel in the image as the center point of the window. Then, the mean and standard deviation of gray values of all the pixels within the window range are calculated. The threshold value of the point is determined by the mean and standard deviation. The connected region segmentation marks each connected region in an image, gives a corresponding threshold, and removes the connected regions exceeding the threshold. The region unsprayed pesticides and the vein part can be removed from the images accurately in this way.

### 2.4. Image Feature Extraction 

In the image of an object, the characteristics that distinguish the object from the other objects are called image features. The image features extracted in present work must reflect the differences between images sprayed with pesticides and those not sprayed. The Gilles spots of the near-infrared microscopic image after image segmentation were determined, and most of them were labeled. Since the number of Gilles spots is correlated with the number of pesticide droplets directly, it was used as one element of the feature vector and was denoted as nmGilles in this work. The darker the color, the higher the concentration of residual pesticides is. Moreover, the uniformity of the color may reflect the characteristics of the area sprayed with pesticides to some degree. In the present study, the first two color moments of the image were extracted and used as the image features. The first-order color moments (mean for short in this study) can reflect the mean value of the overall color of the image, and the second-order color moments (sig for short in this work) reflect the range of the image color distribution. LoG spots is a method that uses the Laplacian of Gaussian (LoG) operator to detect image spots. The number of spots that is greater than a certain threshold was selected as one element of the image feature vector, and it was recorded as nmLoG.

Using the above image feature extraction method, the spray rate feature Ω, color moment features hMean, sMean, sSig, vMean, and vSig, LoG spot feature of nmLoG, and Gilles spot feature of nmGilles were extracted to be features of the images. They are shown in Table 1. Before these digitalized image features were used to build the classification models, they were pretreated by normalization, self-scaling, and regularization methods [28] to eliminate the effect of the difference between the magnitude of image features on model outcomes.

### 2.5. Pattern Recognition of the Images

All classification models of judging whether the vegetable leaves were sprayed by pesticides was built on the basis of the features listed in Table 1. Four common pattern recognition methods of K-nearest neighbors (KNN) [28], naïve Bayes classifier [29], support vector machine (SVM) [30], and artificial neural network of back propagation (BP-ANN) [31] were applied in the present study to build classification models. The accuracy (ACC for short) of the models was evaluated by the following calculation:ACC = Nc/Nv (1)
where Nc is the number of validation samples discriminated correctly, and Nv is the number of all validation samples.

The KNN method “writes” the training samples of known categories into the multidimensional feature space, counts the category of samples with the most K nearest neighbors of each unknown sample, and determines that the unknown sample belong to the category having most near neighbors. The KNN method does not require a training process but requires calculation of the distances between unknown sample and each sample in the training set. K is an odd number that is typically less than 9.

The naïve Bayes classifier [29] is a simple and effective classifier that has been widely used in data mining tasks such as classification and clustering. The naïve Bayesian method assumes that the elements of the feature vector are independent with respect to the category and learns the joint probability distribution of output–input in terms of training set. On the basis of the learned model, the method inputs x (the image features of samples to be discriminated) to obtain the output value that maximizes the posterior probability. 

The core idea of SVM method is to find an optimal hyperplane that separates two classes of samples so that the classification interval of different classes of samples is maximized and the learning strategy of SVM can be reduced to solving a convex quadratic programming problem. The appropriate SVM kernel function can effectively improve the model performance. In addition, different kernel functions map different high-dimensional spaces. So, different SVM classification models were generated by using different kernel function in present work. When an SVM classifier algorithm is developed, the kernel function needs to be selected by considering the complexity of the algorithm and the recognition performance of the algorithm. A linear kernel function is generally chosen when the number of features and the number of samples are close to each other. If the number of features is small and the number of samples is large, a Gaussian radial basis kernel function is generally used.

Back propagation artificial neural net (BP-ANN) is one of the most used artificial neural network models, which has a back propagation error at its core and uses a gradient descent algorithm to adjust the weights and thresholds and subsequently optimize the output. The initial weights of BP-ANN are a random non-zero number. After providing the input and output data matrices, the BP-ANN calculates the inputs and outputs of all implicit layer neurons and output layers. The weights of the hidden layers are modified according to the calculation error, and the inputs and outputs of the hidden and output layers are calculated again until the model training is completed when the error is lower than the given threshold. Present work used one hidden layer and one output. The output of samples sprayed with pesticides and the output unsprayed with pesticides were denoted as 1 and 0, respectively. 

### 2.6. Calculation of Pesticide Degradation Time

Greenhouse cultivation of crops isolates some pests to a certain extent, but the closed environment, stable humidity, and temperature accelerates the breeding of pests and diseases, calling for more pesticide uses. The residues of organophosphates, substituted benzenes, and other pesticides in greenhouse vegetables are all higher than in open-air vegetables. Therefore, it is safe to pick the vegetables sprayed with pesticides when the concentration of residual pesticides on vegetables degrades to the maximum residual safety limit. 

In order to estimate the degradation time of the residual pesticides, the initial concentration of residual pesticides on the vegetables should be known. Present work established a multiple linear regression relationship between the image features listed in Table 1 and the concentration of residual pesticides on the Shanghaiqing foliage sprayed with different concentrations of cypermethrin. The model was used for the prediction of initial concentration of the pesticide C_0_, which was used in the estimation of pesticide degradation time.

The degradation of pesticides in the natural environment can be roughly divided into two steps: primary degradation and secondary degradation. Primary degradation is the structural degradation of the parent compound of the pesticide. In secondary degradation, the products as a result of primary degradation are completely degraded and no longer lead to contamination of crops and the surrounding environment [32]. Most pesticide degradation processes can be expressed in terms of chemical first-order reaction kinetic equations. Therefore, a primary reaction kinetic equation was applied to obtain the degradation equation of pesticides in present study. 

The concentration of pesticides in the natural environment is generally very low, and their degradation dynamics can be described approximately by the following equation:(2)$$-\frac{dc}{dt}=kc$$

Integration of Equation (2) is as follows:(3)$$C_{t}=C_{{}^{0}}e^{\mathrm{kt}}$$
where K is the degradation rate constant, and C_0_ (mg/kg) is the initial concentration of residual pesticides. C_t_ is the concentration of the residual pesticide after a time duration of t days (d). Taking the logarithm of Equation (3), following Equation (4) can be obtained:(4)$$Kt=\ln\frac{C_{0}}{C_{t}}$$

Since the half-life period of *t*_0.5_ corresponds to *C_t_* = 0.5*C*_0_, Equation (4) can be expressed as: (5)$$t_{0.5}=\frac{\ln2}{K}$$
where *t*_0.5_ can be searched from the data bank built in this study and is listed in Appendix A. Substituting *K* = ln2/*t*_0.5_ into Equation (3), the pesticide degradation equation is obtained as the following:(6)$$t=\frac{t_{0.5}}{\ln2}\ln\frac{C_{t}}{C_{{}^{0}}}$$

When the *C_t_* of the above equation is substituted by the threshold of the residual pesticide, which is specified in the national standard, the degradation time that the concentrations of residual pesticide degrade to the specified level can be calculated by Equation (6). The estimated degradation time of pesticides can provide guidance for agricultural producer regarding the right time to harvest crops.

## 3. Experimental Process and Discussion of Results

Pesticides spray quality is usually measured by droplet size (droplet diameter), uniformity of distribution (droplet density), and coverage of the target [33,34]. These metrics are determined generally by the atomization method and the type of spray nozzle [35]. The pesticides examined in this work include cypermethrin and trichlorfon; both are foliar surface insecticides. In the present work, a droplet test card was used to measure the droplet density. The spray quality of several common pesticide spraying methods was inspected and compared to determine the type of fogging device that is suitable for the study. According to the droplet diameter of different fogging methods corresponding to the control object, the liquid force fogging method can produce droplets of 100~300 μm diameter and is more cost effective; therefore, it was selected in this work. 

### 3.1. Sample Preparation and Acquisition of Near-Infrared Microscopic Images

Cypermethrin and trichlorfon with specific concentrations were sprayed on Shanghaiqing samples; the leaves sprayed with the pesticides and the leaves unsprayed with pesticides were collect after standing for 20 h. First, the sludge on the samples was removed. Then, from each leaf sample a section thinner than 0.5 mm was created. Each sample was a rectangular blade approximately 20 mm long and 15 mm wide. The prepared sections were placed on a glass slide on which another slide was placed and compacted, so that the Shanghaiqing leaves between the two slides were flat. The leaf slides were loaded into the sample holder (201) in the NIR microscopy imaging system, then the switch 113 was opened to supply power to the system. The focal length was adjusted through the micro-lens (202) to get a clear microscopic image of the leaf, and wavelength of the complementary light by (113) was switched after getting a clear image. The CMOS industrial camera (203) was operated at the intelligent terminal (300) to take microscopic images and obtain near-infrared microscopic images of the leaves of Shanghaiqing.

### 3.2. Design of the Concentration of Sprayed Pesticide Solutions

Present work selected the spraying concentration range of the pesticides on the basis of the type of the pesticides and their conventional spraying concentration. 

Cypermethrin is a moderately toxic pyrethroid insecticide that is insoluble in water [36]. In this study, cypermethrin powder was dissolved in acetone and configured to a concentration of 100 g/L in acetone solution to prepare the mother liquor of cypermethrin. Per the commonly used concentration of cypermethrin, the mother liquor was then diluted 1000 times with pure water, and the diluted cypermethrin solution was sprayed to Shanghaiqing to prepare samples sprayed with cypermethrin. In addition, two cypermethrin solutions were prepared by diluting the mother liquor 2000 times and 4000 times, respectively, to examine the detectability of different concentrations of cypermethrin residues attached to the leaves of Shanghaiqing. Trichlorfon is an organophosphate insecticide with high insecticidal activity and acute toxicity [37]. Trichlorfon was configured as an aqueous solution with a concentration of 100 g/L in pure water to prepare the mother liquor of trichlorfon. Then, the mother liquor was diluted to 100 times with pure water. The diluted trichlorfon solution was sprayed to Shanghaiqing in this study to prepare samples sprayed with trichlorfon.

The two types of pesticides solutions were loaded into a hydraulic atomizer and sprayed onto Shanghaiqing and droplet test cards. The spray density given by Image J 1.4.3 software represents the number of droplets per unit area, and the droplet density can be used to calculate the coefficient of variation (CV), which was applied to evaluate the uniformity of droplet distribution. The calculation formula of CV is as follows: (7)$$CV=\frac{S}{\bar{X}},S=\sqrt{\frac{\sum_{i=1}^{n}{\left(X_{i}-\bar{X}\right)}^{2}}{n-1}}$$
where *S* is the standard deviation of droplet density, $X_{i}$ is the spray density on each droplet test card, $\bar{X}$ is the average of spray density on all droplet test cards, and *n* is the total number of all droplet test cards.

The requirements of the industry standard for motorized sprayers under the three modes of conventional volume spraying, low volume spraying, and ultra-low volume spraying are ≤50%, ≤50%, and ≤70%, respectively. In the present work, the uniformity of droplet distribution must meet the requirements under conventional volume spraying, i.e., the coefficient of variation (CV) is less than 50%. 

### 3.3. The Sample Sets and Their Usage

From a statistics perspective, the size of positive and negative groups should not differ too much. On the basis of this idea, the number of samples in negative groups (unsprayed with pesticides) were designed to be approximately half of that in positive group (sprayed with pesticides). Five sample sets were constructed in this work to build classification models for discriminating Shanghaiqing samples sprayed with pesticides. The composition of the five sample sets are shown in Figure 4. 

Set 1 consists of 115 samples unsprayed with pesticides and 212 samples sprayed with the trichlorfon solution of 1 g/mL concentration, which were prepared in 3.1. This set was used to establish and validate classification models for discriminating samples sprayed with trichlorfon. 

Set 2 consists of 49 samples unsprayed with pesticide and 107 samples sprayed with the cypermethrin solution of 0.1 g/mL, which were prepared in 3.1. This set was used to establish and validate the classification models for discriminating samples sprayed with cypermethrin. 

Since the sizes of the sample unsprayed with pesticides in Set 1 and Set 2 are approximately half of those sprayed with pesticides, in order to increase the number of samples unsprayed with pesticides, the unsprayed samples in Set 1 and Set 2 were combined to form a total of 164 samples unsprayed with pesticides. These 164 samples and the 212 samples of Set 1 sprayed with trichlorfon formed Set 3. It was used to build and validate classification models for discriminating samples sprayed with trichlorfon. The results of the models were compared with those of the models built on Set 1 to examine the effect of the difference in the number of samples unsprayed with pesticides on the models’ results. 

The 164 samples unsprayed with pesticides were combined with the 107 samples sprayed with cypermethrin of Set 2 to form Set 4, which was used to build and validate the classification models of discriminating samples sprayed with cypermethrin. The results of the models built on Set 4 were compared with those of the models built on Set 2 to examine the effect of the difference in the number of samples unsprayed with pesticides on the models’ results. 

Furthermore, adding the 212 samples sprayed with trichlorfon in Set 1 and the 107 samples sprayed with cypermethrin in Set 2 together, 319 samples sprayed with pesticides were obtained. These 319 samples and the 164 samples consisting of pesticide-free samples of Set 1 and Set 2 constituted Set 5, which was used to establish and validate the classification models for discriminating samples sprayed with pesticides (i.e., without distinguishing which pesticide was sprayed).

In order to inspect the sprayed pesticide concentration on the accuracy of the SVM model, three sample sets of Set 6, Set 7, and Set 8 were designed, and their composition is shown in Figure 5. 

As shown in Figure 5, the cypermethrin mother liquor of 100 g/L prepared in 3.1 was diluted 1000, 2000, and 4000 times with pure water, respectively. Three groups of samples were prepared. Each group had 50 Shanghaiqing samples which were sprayed with the diluted cypermethrin solutions at one of the above three concentrations and had 30 Shanghaiqing samples sprayed with pure water. Sample Set 6, Set 7, and Set 8 were ranked from the highest to the lowest concentration of cypermethrin.

### 3.4. Discrimination Results of Pesticides Residue

First, the captured true color images in jpg format with 24-bit depth were grayed out before they were used to build models. The greyscale image was processed using a Gaussian filter template of size 5 × 5 and σ = 1. A Retinex algorithm with Gaussian surround scale c = 250 was used to perform a light uniformity operation on the denoised image. Image segmentation was performed by using the Niblack local thresholding segmentation method with a sliding window of 251 and k = −0.1. The leaf vein part of the image was segmented using the connected domain thresholding method. A 5 × 5 disc-type mask was applied to close the image to obtain the image of the pesticide spraying area.

For the sample sets described in 3.3, their image features shown in Table 1 of 2.4 were extracted as the basis of building the classification models.

#### 3.4.1. Discrimination Results of Classification Models Based on Set 1 and Set 2

On the basis of the manual labeling of the images, 1 and 0 were applied to denote spraying (positive) and non-spraying (negative) images, respectively. The final image feature matrixes of Set 1 and Set 2 were formed after labeling the classes. Different classification models were established by the four methods described in 2.5 in terms of the image feature matrix shown in Table 1 of Section 2.4. The accuracy of internal validation set of these models is listed in Table 2. 

As shown in Table 2, the SVM models with linear or RBF kernel functions provide the highest or the second highest ACC regardless of whether Shanghaiqing leaves are sprayed with trichlorfon or cypermethrin when the image features are pretreated by regularization. In addition, the discriminative accuracy of the SVM models built on the image features pretreated by other methods are higher than 70%, and the value is higher than the lowest ACC of the models established by KNN, Bayes, and BP-ANN methods.

The BP-ANN model with a training time of 15 has the highest ACC of 97.98% when the image features are pre-processed with the standardization or self-scaling. This result is slightly better than the accuracy of 96.96% given by the SVM model with RBF kernel function and regularization pretreatment. However, when the BP-ANN model is built on preprocessing the data with the regularization method, its accuracy for validation samples of Set 1 is only 52.53%. 

As for the classification models built by KNN and naïve Bayes methods, their highest accuracies are not all as high as SVM models, while their lowest accuracy are all lower than SVM models. The lowest ACC of Set 1 given by naïve Bayesian is only 21.21%, which is obtained by the pretreatment method of regulation, while the highest ACC of Set 1 given by naïve Bayesian is 92.36%, which is obtained by pretreatment methods of standardization or self-scaling. The results show that pretreatment methods impact discrimination accuracy of naïve Bayesian greatly for Set 1, and naïve Bayesian method is not robust. In summary, the accuracy and robustness of the models established by SVM with RBF kernel function are better than those in the models built by the other three methods. Therefore, the SVM method was subsequently used to discriminate samples in the mixed sample sets of Set 3~Set 5.

Comparing the results of the SVM models built on Set 1 (SVM-Set 1 for abbreviation) with the SVM models built on Set 2 (SVM-Set 2 for abbreviation), it is found that the accuracy of SVM-Set 1 is 13~17% higher than that of SVM-Set 2. The concentration of the pesticide sprayed on Shanghaiqing in Set 1 is 1 g/L, which is ten times that of the concentration of the pesticides sprayed on Shanghaiqing in Set 2 (0.1 g/L). Trichlorfon and cypermethrin contain different groups. The near-infrared absorption intensity and the spraying concentration of the two types of pesticides are different. Thus, the quality of near-infrared microscopic images of samples sprayed with cypermethrin would not be as good as that of the images of samples sprayed with trichlorfon. This might be the reason why SVM-Set 1 performs much better than SVM-Set 2. In addition, the number of samples of Set 1 is approximately two times that of the samples in Set 2. This might be another reason that the accuracy of SVM-Set 1 is higher than that of SVM-Set 2.

Limited by the length of this article, only false positive rate (FPR, i.e., the probability that the classification model determines the samples unsprayed with pesticides to be samples of sprayed with pesticides) corresponding to the highest ACC of each recognition method was listed in the fifth column of Table 2. The detailed information of false negative rate, true positive rate, and true negative rate can be found in the Appendix A. As shown in Table 2, the FPR corresponding to the highest ACC of SVM is between 3.34% (for Set 1) and 13.34% (for Set 2); the FPRs of other methods are also in this range, and the values of FPR are at acceptable levels.

#### 3.4.2. Discriminant Results of Classification Models Based on the Mixed Sample Set 3, Set 4, and Set 5

The results of SVM models built on Set 3~Set 5 are shown in Table 3.

Per Table 3, the accuracy of SVM classification models built on Set 3 (SVM-Set 3 for short) is in the region of 87.61~93.80%, while the accuracy of SVM-Set 1 is in the range of 83.20~96.96%. The SVM-Set 3 models using the linear kernel function always have the highest or the second highest discriminative accuracy of 93.80% and 92.92%, respectively, which are lower than the highest accuracy of 96.96% given by SVM-Set 1. The result shows that increasing the number of samples unsprayed by pesticides does not improve the discrimination accuracy of the SVM models for assessing whether the samples are sprayed with Trichlorfon. 

The SVM classification models built on Set 4 (SVM-Set 4 for short) with RBF kernel function built on the image features pretreated with self-scaling and standardization methods have the highest discrimination accuracy of 72.47%. It is lower than the accuracy of 79.10% given by the SVM-Set 2 in Table 2, which has a linear or RBF kernel function and was built on the image features pretreated by regularization. The result indicates that adding the samples unsprayed by pesticides of Set 2 reduces the discrimination accuracy of the SVM models for assessing whether the samples are sprayed with cypermethrin. 

The SVM discriminant models built on Set 5 (SVM-Set 5 for short) do not consider the variety of sprayed pesticides and simply assess whether the samples were sprayed with pesticides. The discrimination accuracies of SVM-Set5 fluctuate between 77.24% and 86.20%. The accuracies are significantly lower than the discriminant accuracies of the SVM-Set 1 model in Table 2 (83.2–96.96%) but are higher than the discriminant accuracy of the SVM-Set 2 model in Table 2 (70.83~79.16%): approximately 7%. This result indicates that the discrimination results of the SVM classification models are closely related to the data structure and complexity. 

In Set 5, the number of samples sprayed with trichlorfon of 1 g/L concentration is approximately two times that of samples sprayed with cypermethrin of 0.1 g/L, i.e., the samples sprayed with pesticide at higher concentration are approximately twice that of the samples sprayed with pesticide at a lower concentration, and the sprayed pesticide concentration has two levels which are different by one magnitude. The samples in Set 5 are more complicated than Set 1~Set 4. Therefore, the accuracy of SVM-Set 5 is higher those of SVM-Set 2 and SVM-Set 4 whose samples were sprayed with low-concentration pesticide, and lower than those of SVM-Set 1 and SVM-Set 3 whose samples were sprayed with high-concentration pesticide.

### 3.5. Effect of Pesticide Concentration on the Classification Models

As shown in Figure 5, 150 samples sprayed with cypermethrin at three concentration levels and 30 Shanghaiqing leave samples sprayed with water were prepared to investigate the effect of pesticides concentration on the classification models. One of the local threshold segmentation images of each group of Figure 5 with the addition of connected domain segmentation is shown in Figure 6. As shown in Figure 6, the higher the pesticide spraying concentration, the better the image segmentation of the cypermethrin droplets and the better the image of Shanghaiqing leaves themselves. For the combination of cypermethrin and pure water spraying groups with 1000×, 2000×, and 4000× dilution, respectively, the RBF kernel SVM model was used to identify whether cypermethrin was sprayed on Shanghaiqing, and the ratio of training set to validation set was 3:2. The discrimination accuracy of the SVM classification models based on the three sample groups is shown in Table 4.

It can be seen from Table 4 that the SVM models built on the images pre-processed by regularization have the best classification results for the three groups of samples. The highest and the lowest accuracy occur at the cypermethrin concentrations of 0.1 g/L and 0.025 g/L, respectively. When the concentration of cypermethrin is 0.05 g/L, the model can maintain a high discrimination accuracy of approximately 80%. When the spraying concentration of cypermethrin reduces to 0.025 g/L, the accuracy of the model decreases significantly, which is less than 70% in the best case. This indicates that the accuracy of the SVM model is positively correlated with the concentration of cypermethrin sprayed on the leaves of Shanghaiqing.

### 3.6. Estimating Results of Pesticide Degradation Time

On the basis of the highly independent characteristics of near-infrared microscopic images of Shanghaiqing sprayed with different concentrations of cypermethrin of Set 6~Set 8, a multiple linear regression relationship between these characteristics and the pesticide spraying concentration was established as follows:(8)$$\begin{array}{c} y=0.0599-0.0205\times \Omega +0.5713\times hMean-0.3533\times sMean\\ -0.0496\times sSig-0.0686\times vMean+0.0543\times vSig \end{array}$$

The statistics index F = 7.22, which is greater than F0.05(6, ∞) = 2.72, and the *p*-value of the Equation (8) is 0.0017, which is smaller than 0.01. Therefore, the regression Equation (8) is statistically significant, and there is a linear relationship between cypermethrin spraying content y and the six image features. Using Equation (8) for quantitative prediction of cypermethrin concentration on Shanghaiqing, the actual time required for pesticide degradation in the samples with the maximum residue limit specified in the national standard (Appendix A) can be calculated by substituting the *C*_0_ (y predicted by Equation (8)) and the natural environmental degradation half-life of the pesticide (*t*_0.5_ = 1d for trichlorfon-soil and *t*_0.5_ = 2.8d in cypermethrin-soil) into Equation (6) in Section 2.6.

## 4. Conclusions

The present work shows that it is feasible to discriminate vegetables sprayed with pesticides on the basis of NIR microimaging images and CV technology. Based on the eight features of Table 1 extracted from the near-infrared microscopic images of Shanghaiqing, it is seen that the SVM method with linear or RBF kernel function can build robust classification models, which can provide discrimination accuracy of 96.96% for the Shanghaiqing foliage sprayed with trichlorfon solution of 1 g/L and the accuracy of 79.16~84.37% for discriminating the Shanghaiqing foliage sprayed with cypermethrin solution of 0.1 g/L. With the decrease of the concentration of sprayed pesticides, the discrimination accuracy of the SVM models built on the near-infrared microscopic images of Shanghaiqing foliage decreases. The concentration of sprayed pesticides and the complexity of sample sets have significant impacts on the classification results.

The pesticide residue rapid detection system developed in this work offers guidance for agricultural producers on harvesting agricultural products at an appropriate time while ensuring that the pesticide residues do not exceed the national limit. It has the advantages of high portability, fast detection, low professional requirement for the operator, and low cost of equipment. This study provides support for the primary screening and prognosis of pesticide residues in the grassroots market. The testing conditions of the current prototype are laboratory environment, and more field tests can be conducted in the future.

## Figures and Tables

**Figure 1 sensors-23-00983-f001:**
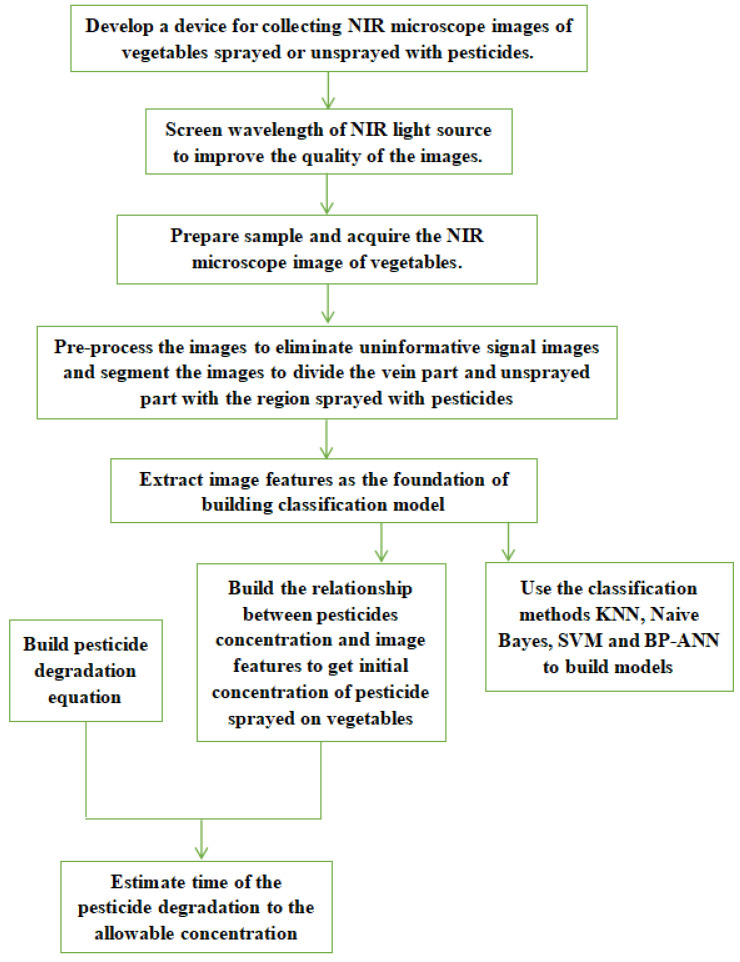
The research roadmap of using NIR microscopic images and CV technology to detect pesticide residue on green vegetables.

**Figure 2 sensors-23-00983-f002:**
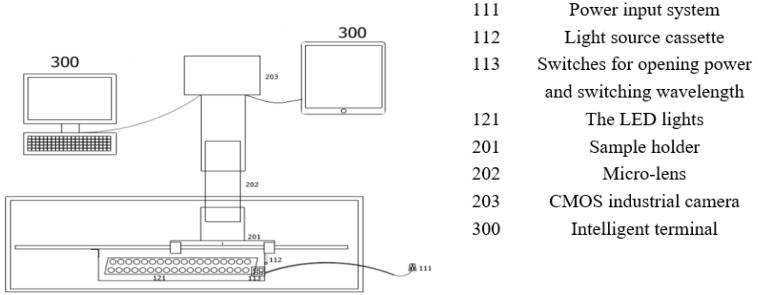
Structure sketch of near-infrared microscopic imaging system.

**Figure 3 sensors-23-00983-f003:**
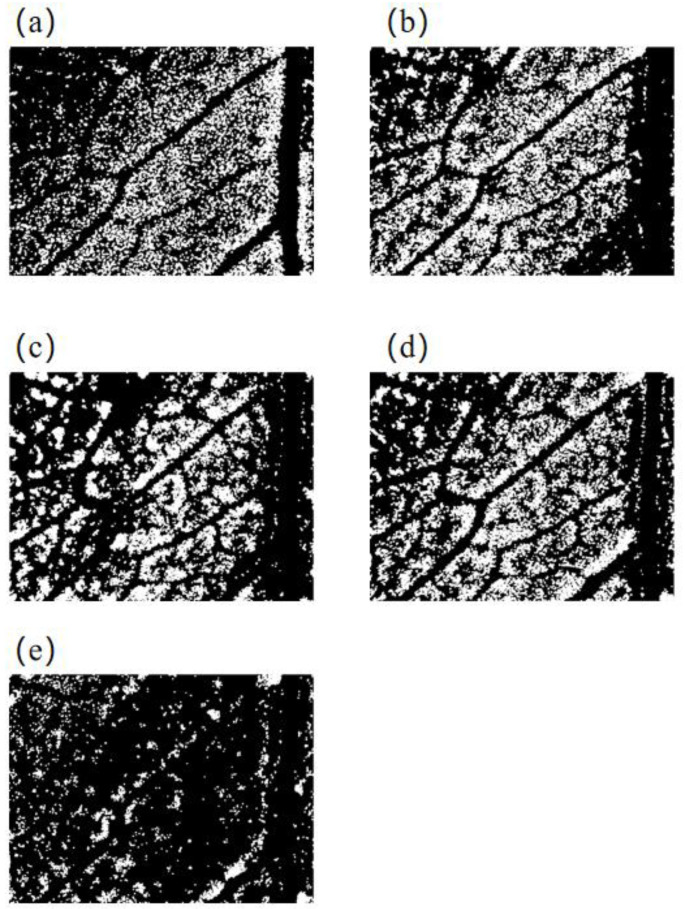
Segmentation results of microscopic images of Shanghaiqing leaf sprayed by cypermethrin with 0.1 g/L taken under the light sources at different wavelengths. (**a**–**e**) are infrared microscopic image segmentation of Shanghaiqing leaf taken with the light source at 430 nm, 470 nm, 560 nm, 660 nm, and 935 nm, respectively.

**Figure 4 sensors-23-00983-f004:**
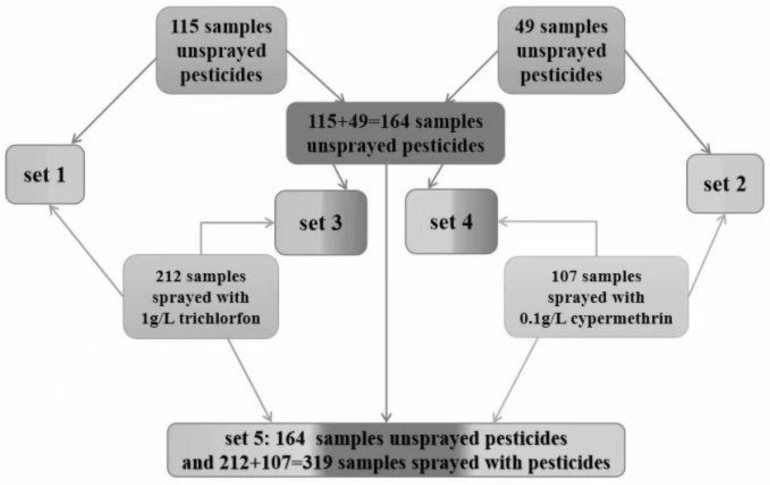
The constitution of samples sets used to build classification models for discriminating Shanghaiqing sprayed with trichlorfon or cypermethrin.

**Figure 5 sensors-23-00983-f005:**
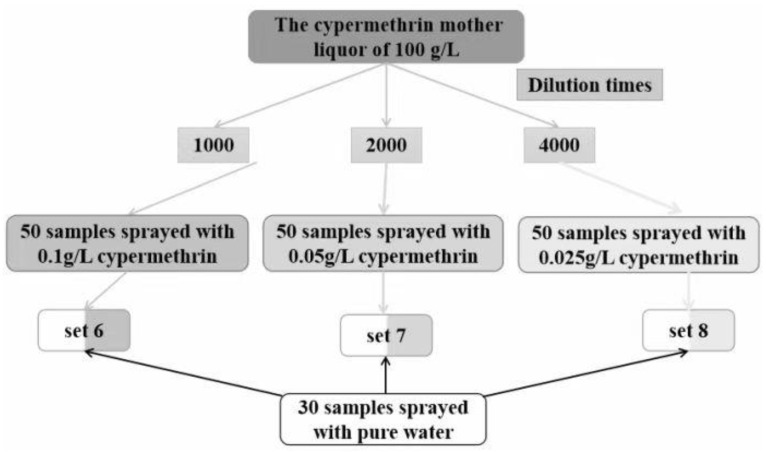
The composition of the three sets used to examine the effect of concentration of cypermethrin on the accuracy of SVM classification models.

**Figure 6 sensors-23-00983-f006:**
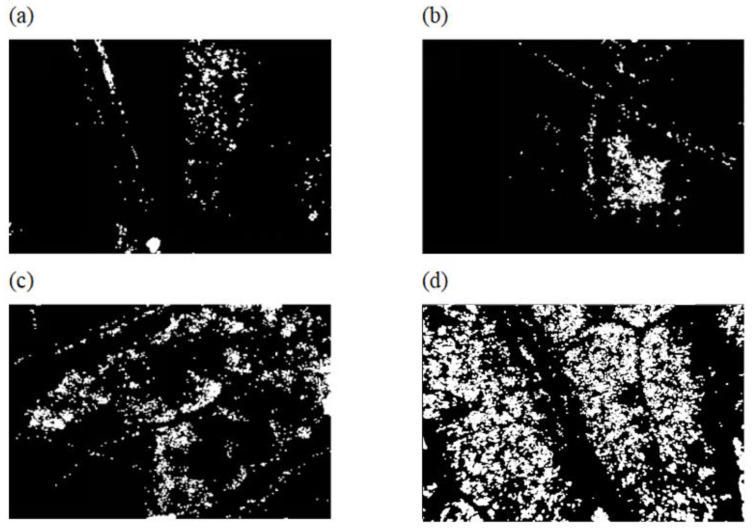
Linear scale 1:400. Segmentation results of microscopic images of Shanghaiqing leaf sprayed by different solutions. (**a**–**d**) are infrared microscopic image segmentation of Shanghaiqing leaf sprayed with pure water, sprayed with 0.025 g/L cypermethrin, sprayed with 0.05 g/L cypermethrin, and sprayed with 0.1 g/L cypermethrin, respectively.

**Table 1 sensors-23-00983-t001:** Microscopic near-infrared image features of Shanghaiqing samples.

Feature Type	Feature Code	Samples without Pesticide Residues	Sample Sprayed with Pesticides
Spraying rate characteristics	Ω	0.0997~0.2868	2.5 × 10^−5^~0.0633
Color moment characteristics	^h^ Mean	0.7835~0.7980	0.8244~0.7875
	^s^ Mean	0.5907~0.6604	0.5078~0.6491
	^s^ Sig	0.0104~0.1607	0.0224~0.0115
	^v^ Mean	0.3552~0.8926	0.6957~0.6686
	^v^ Sig	0.5365~0.1858	0.1339~0.1062
LoG spot features	nmLoG	3~215	1~256
Gilles spot features	nmGilles	1~13	1~13

Note: Prefix “^h^” component represents hue, prefix “^s^” component represents saturation, and prefix “^v^” component represents value.

**Table 2 sensors-23-00983-t002:** Results of different classification models built on Set 1 and Set 2.

Pesticide Variety/Concentration	Sample Set	Classification Method	Scope of ACC (%)	FPR (%)	Pretreatment Method	Parameters Used in Classification Methods
Trichlorfon/1 g/L	Set 1	KNN	56.56–92.92	13.33	Standardization or self-scaling	K = 3
		Naïve Bayes	21.21–92.36	13.21	Standardization or self-scaling	/
		SVM	83.20–**96.96**	3.34	Regularization	Linear or RBF kernel function
		BP-ANN	52.53–**97.98**	3.34	Self-scaling	Times of training = 15
Cypermethrin/0.1 g/L	Set 2	KNN	66.67–79.16	13.33	Standardization or self-scaling	K = 1
		Naïve Bayes	64.58–77.77	11.54	Standardization or self-scaling	/
		SVM	70.83–**79.16**	13.34	Regularization	RBF kernel function
		BP-ANN	68.75–75.00	6.67	Standardization or self-scaling	Times of training = 15

FPR(Detailed in Appendix A) is false positive rate corresponding to the highest ACC; pretreatment methods and the parameter used in the corresponding method are those that resulted in the largest ACC; for Bayes method, the largest ACC is obtained by dividing ratio of number of calibration samples to number of validation samples = 3:2, while the ratio of the other three methods is 7:3. The pretreatment method is the method when the maximum ACC is reached. The bold in the table is the result with high accuracy and will be used for the discussion below.

**Table 3 sensors-23-00983-t003:** Results of SVM classification models built on the mixed sample sets of Set 3, Set 4, and Set 5.

Pesticide Variety/Spray Concentration	Sample Set	Pre-Processing Method	Nt/Nv	ACC (%)		
				Linear Kernel	Polynomial Kernel	RBF Kernel
Trichlorfon/ 1 g/L	Set 3	Standardization	263:113	92.92	89.38	87.61
		Self-scaling	263:113	92.92	89.38	87.61
		Regularization	263:113	**93.80**	90.26	**93.80**
Cypermethrin/ 0.1 g/L	Set 4	Standardization	162:109	71.55	71.55	**72.47**
		Self-scaling	162:109	71.55	71.55	**72.47**
		Regularization	162:109	63.30	65.13	67.88
Trichlorfon/ 1 g/L and Cypermethrin/ 0.1 g/L	Set 5	Standardization	338:145	77.24	82.75	82.06
		Self-scaling	338:145	77.24	82.75	82.06
		Regularization	338:145	**86.20**	82.75	83.44

Note: Nt is the number of samples in the training set; Nv is the number of samples in the validation set; and ACC is the discrimination accuracy rate of validation samples.The bold in the table is the result with high accuracy and will be used for the discussion below.

**Table 4 sensors-23-00983-t004:** Results of SVM classification models with RBF kernel function for discriminating Shanghaiqing sprayed with different concentration cypermethrin.

No.	Pre-Processing Method	Nt/Nv	ACC (%)		
			Concentration of Cypermethrin		
			0.1 g/L	0.05 g/L	0.025 g/L
1	Standardization	48:32	75.00	78.12	56.25
2	Self-scaling	48:32	75.00	78.12	56.25
3	Regularization	48:32	84.37	81.25	68.75

Note: Nt is the number of samples in the training set; Nv is the number of validation samples; and ACC is the accuracy rate of validation set.

## Data Availability

Detailed data supporting the results of the report can be obtained from the charts in this paper and Appendix A.

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
