# Peer review of "Study on Rapid Detection of Pesticide Residues in Shanghaiqing Based on Analyzing Near-Infrared Microscopic Images"

_sensors, 2023, doi:10.3390/s23020983_

Round 1
Reviewer 1 Report
line 37: effective and rapid methods of testing pesticides are essential to detect food safety but not for "reducing risks of overuse pesticide."
The authors also need to pay attention to the formatting issues. For example, line 73, Sankar K et al. ... delete "K" here.
I am not the expert of the subject, so not many suggestions could be given.
Author Response
Response to Reviewer 1 Comments
We are very grateful for the your comments on our manuscript.
After carefully reading the comments, we revised the manuscript and invited a friend who is good at English to check language and grammar of the revised manuscript. All the modifications are marked in the text with yellow highlight. We hope that the correction will meet with approval.
The Point-by-point responses to the reviewers’ comments are as followings.
Point 1: line 37: effective and rapid methods of testing pesticides are essential to detect food safety but not for "reducing risks of overuse pesticide."
Response 1: Thank you very much for this suggestion, we had revised this sentence in line 36-37 of the revised manuscript.
Point 2: The authors also need to pay attention to the formatting issues. For example, line 73, Sankar K et al. ... delete "K" here.
Response 2: Yes, we deleted the letter in line 78 of the revised manuscript.

Reviewer 2 Report
IR spectroscopy is a powerful technique for biological sample's analysis. Pesticides are critical for agricultural productivity; however, they are obnoxious contaminants and toxic substances. Hence, require powerful methods for detection and prevention of its entry in soil, water, food, and the environment. Authors have performed a key study to achieve these goals in Brassica species.
Few suggestions to improve the quality:
Please check the scientific name and write correctly.
Line 108: ‘are also be different’, grammatically wrong
Figure 1. Elaborate the caption of this figure.
The same could be applied for figures 2, 3, and 4.
For flowcharts, do not fill in the boxes with colors; or use light colors to fill up. The non-uniform or too bright/too dark coloring is distracting to the readers.
For Figure 5. Add scale/size bars. Scale bars are missing.
The authors need to improve the introduction and discussion of the study, where authors can compare other aptamer-based biosensors used for pesticide analysis. For example—see https://doi.org/10.1016/j.teac.2022.e00184. I recommend some of the latest studies on this topic by authors of this Trends review, they have demonstrated aptamers are useful for analysis of fenitrothion, fipronil, malathion, and diazinon. Authors of present study can discuss these studies along with IR spectroscopy applications.
Conclusions could be rewritten with key findings and future implications of the study.
If possible, authors could include statistical significance of the observations regarding microscopic images of leaves.
Author Response
Response to Reviewer 2 Comments
We are very grateful for the your comments on our manuscript.
After carefully reading the comments, we revised the manuscript and invited a friend who is good at English to check language and grammar of the revised manuscript. All the modifications are marked in the text with yellow highlight. We hope that the correction will meet with approval.
The Point-by-point responses to the reviewers’ comments are as followings.
Point 1: Please check the scientific name and write correctly.
Response 1: Thank you for reminding it. We changed the scientific name of the vegetable to Shanghaiqing, which is a kind of Chinese green vegetables. Shanghaiqing is its Chinese Phonetic Transcription. And we highlight the word "Shanghaiqing" in the text.
Point 2: Line 108: ‘are also be different’, grammatically wrong
Response 2: Thank you for pointing out this mistake. We already corrected it in line 118 of the revised manuscript.
Point 3: Figure 1. Elaborate the caption of this figure. The same could be applied for figures 2, 3, and 4.
Response 3: Thank you. We have elaborated the caption of the figures. And we highlight the modified statement in line 145-146, line 170, line 419-420 and line 453-454.
Point 4:For flowcharts, do not fill in the boxes with colors; or use light colors to fill up. The non-uniform or too bright/too dark coloring is distracting to the readers.
Response 4: Thank you for this kindly suggestion. We have changed the figure 4 and figure 5 by filling the box with white and gray at different levels.
Point 5:For Figure 5. Add scale/size bars. Scale bars are missing.
Response 5: Thanks for this reminding. The scale bars were added in Figure 6 of the revised manuscript in line 583.
Point 6:The authors need to improve the introduction and discussion of the study, where authors can compare other aptamer-based biosensors used for pesticide analysis. For example—see https://doi.org/10.1016/j.teac.2022.e00184. I recommend some of the latest studies on this topic by authors of this Trends review, they have demonstrated aptamers are useful for analysis of fenitrothion, fipronil, malathion, and diazinon. Authors of present study can discuss these studies along with IR spectroscopy applications.
Response 6: Thanks for this reminding. We cited the reference in line 63-65 of the revised manuscript.
Point 7:Conclusions could be rewritten with key findings and future implications of the study. If possible, authors could include statistical significance of the observations regarding microscopic images of leaves.
Response 7: Thank you for this comment. We rewrote the conclusions and deleted some words about research results. The first paragraph introduces key findings of this study, and the second paragraph states the significance of this work and its effect in agriculture field.

Reviewer 3 Report
This work contains a proposal for a pesticide detection method based on near-infrared imaging at a suitable wavelength, followed by image processing, segmentation, and pattern recognition. The authors claim that this method offers rapid and flexible detection of pesticides, including the possibility of in situ applications, given the simplicity of the required scientific equipment. The problem is of interest and the implementation of the image analysis procedures is solid.
Aside from some instances of poor or unclear writing, my two main concerns with the article are the lack of details about some fundamental aspects of the methodology and the inclusion of brief comments on kinetic aspects, whose interest to the general study I do not understand. I will proceed to enumerate the specific points that I would like the authors to address before recommending the article for publication:
1. The technique is apparently based on bright-field microscopy, making use of the differences in IR absorbance of the pesticides and the leaves. However, the only image provided (Figure 5) looks like a dark-field technique, with a black background and the number of bright spots increasing with increasing concentration. Please, clarify this point.
2. There is no equivalent to Figure 5 for the case of samples sprayed by trichlorfon.
3. In Figure 5a, it seems that water-sprayed leaves present similar images to those of leaves sprayed with low concentrations of pesticide. Did the authors consider the possibility of false positives?
4. An important aspect of the technique is its selection of a suitable operating wavelength. However, only the central wavelength (935 nm) of the chosen LED source is provided. I ask the authors to provide information about this light source, especially concerning its spectral range/degree of monochromaticity.
5. The proposed method requires sample preparation, although a quite simple one. How does this compare to other infrared methods, such as ATR, which are increasingly being used to study materials in the near-infrared range without sample preparation? The authors cite in [21] a reference to FTIR microspectroscopic techniques, but do not discuss them.
6. The authors claim on Page 5 that the leaves do not have an absorption peak at 935 nm, whereas cypermethrin and trichlorfon do. They also claim on Page 13 that the infrared absorption intensity of cypermethrin is weaker than that of trichlorfon at this wavelength. Please, provide references for these claims.
7. In relation to the previous point: is this method generalizable to other pesticides? The authors have not investigated what overtone band is responsible for absorption at the wavelength they chose.
8. In my opinion, Sections 2.6 and 3.6, concerning degradation kinetics of the pesticides, do not add anything significant to the manuscript, and can be omitted completely without compromising the integrity of the article.
9. I suggest that the authors refer more precisely to “near-infrared imaging”, rather than the more general term “infrared”.
10. Section 3 is not properly a Results & Discussion section, as half of it corresponds to a description of the experiment. The real results start on Page 12, not 9.
11. I am definitely not an expert on taxonomy, but it seems that Brassica campestris is a term in disuse, in favor of Brassica rapa. What is the reason for the authors to keep that name?
Author Response
Response to Reviewer 3 Comments
We are very grateful for the your comments on our manuscript.
After carefully reading the comments, we revised the manuscript and invited a friend who is good at English to check language and grammar of the revised manuscript. All the modifications are marked in the text with yellow highlight. We hope that the correction will meet with approval.
The Point-by-point responses to the reviewers’ comments are as followings.
Point 1: The technique is apparently based on bright-field microscopy, making use of the differences in IR absorbance of the pesticides and the leaves. However, the only image provided (Figure 5) looks like a dark-field technique, with a black background and the number of bright spots increasing with increasing concentration. Please, clarify this point.
Response 1: Thank you for this comment. The images used in this study are color in initial. However, they were pretreated into white and black by greying. Therefore, the images in Figure 5 were not color but black and white. We revised the contents in line 224-227 of the revised manuscript to clarify this point.
Point 2: There is no equivalent to Figure 5 for the case of samples sprayed by trichlorfon.
Response 2: For the two kinds of pesticide studied in this work, the sprayed concentration of cypermethrin is one order of magnitude lower than that of trichlorfon. The samples sprayed with cypermethrin are more difficult to be discriminated than those sprayed with trichlorfon. Therefore, we prepared samples sprayed with cypermethrin at different concentration levels to study the effect of pesticide’s concentration on the accuracy of classification models. Only the images of the samples sprayed with cypermethrin were taken in this study. The samples sprayed with trichlorfon at different concentration levels were not prepared, then their images were not provided for comparison in Figure 5 (the figure is Figure 6 in the revised manuscript).
Point 3: In Figure 5a, it seems that water-sprayed leaves present similar images to those of leaves sprayed with low concentrations of pesticide. Did the authors consider the possibility of false positives?
Response 3: Yes, the image of the sample sprayed with low concentration of pesticide is like the image sprayed with water. It is possible that the classification models discriminate the samples sprayed with water to be those sprayed with pesticide. Limited by the length of the article, we listed the related results in the appendix, and only introduced the false positive rate (FPR) in table 2 and discussed the results of FPR in lines 523-530 of revised manuscript.
Point 4: An important aspect of the technique is its selection of a suitable operating wavelength. However, only the central wavelength (935 nm) of the chosen LED source is provided. I ask the authors to provide information about this light source, especially concerning its spectral range/degree of monochromaticity.
Response 4: In line 194-199 of the revised manuscript, the reason of choosing the wavelength of 935nm was explained. We also added Figure 3 to show the spectra of Shanghaiqing, trichlorfon and cypermethrin. According to this figure, the two pesticides used in this study have strong or stronger absorbance peaks at 935 nm but Shanghaiqing does not have obvious absorbance at the wavelength.
Point 5: The proposed method requires sample preparation, although a quite simple one. How does this compare to other infrared methods, such as ATR, which are increasingly being used to study materials in the near-infrared range without sample preparation? The authors cite in [21] a reference to FTIR micro-spectroscopic techniques, but do not discuss them.
Response 5: For NIR spectroscopy technique, whether sample treatment is required depends on sample characteristics but not NIR methods (instruments). For example, samples such as grains and flour do not need to be pretreated. But for leave samples, the simple compressing pretreatment is necessary to obtain high quality spectra and to reduce the complexity of image processing. After tablet compression processing, the captured leave images are on the same plane, which will chalk up good shooting effect and high data repeatability.
In line 107-112 of the revised manuscript, we added the discussion on FTIR micro-spectroscopic techniques.
Point 6: The authors claim on Page 5 that the leaves do not have an absorption peak at 935 nm, whereas cypermethrin and trichlorfon do. They also claim on Page 13 that the infrared absorption intensity of cypermethrin is weaker than that of trichlorfon at this wavelength. Please, provide references for these claims.
Response 6: We added Figure 3 in the revised manuscript to be the references of these claims. The detail discussion on this point is also added in line 200-205 of the revised manuscript.
Point 7: In relation to the previous point: is this method generalizable to other pesticides? The authors have not investigated what overtone band is responsible for absorption at the wavelength they chose.
Response 7: The methodology including the device developed in this study can be generalized to other pesticides. However, the wavelength of the light source used in this study might not be very suitable for other pesticides. In line 200-202 of the revised manuscript, we discussed the overtone bond responsible for the absorption at 935 nm.
Point 8: In my opinion, Sections 2.6 and 3.6, concerning degradation kinetics of the pesticides, do not add anything significant to the manuscript, and can be omitted completely without compromising the integrity of the article.
Response 8: We are sorry for that we did not state the aims of this work clearly in the original manuscript. It makes the reviewer do not think the contents of 2.6 and 3.6 have significance to the manuscript. The work of 2.6 and 3.6 wants to provide support for agricultural producers to harvest crops at an approximate time. This is one of the aims of this study. We hope to keep the contents in the article. In the first sentence of the abstract and line 36-37 of the revised manuscript, we explained the aim of this work and the importance of providing support for guiding agricultural producers to harvest crops at suitable time to ensure the pesticide residue on crops be at an allowable level.
Point 9: I suggest that the authors refer more precisely to “near-infrared imaging”, rather than the more general term “infrared”.
Response 9: Thank you for your kindly reminding, the wavelength of the light source is really in near-infrared spectral range. We have corrected all “infrared” to be “near-infrared” or “NIR”.
Point 10: Section 3 is not properly a Results & Discussion section, as half of it corresponds to a description of the experiment. The real results start on Page 12, not 9
Response 10: We changed the title of section 3 to Experiment process and discussion on results (line 355) in the revised manuscript.
Point 11: I am definitely not an expert on taxonomy, but it seems that Brassica campestris is a term in disuse, in favor of Brassica rapa. What is the reason for the authors to keep that name?
Response 11: We are sorry for using a confusing name to describe the vegetable used in this study. We changed the name of the vegetable to Shanghaiqing, a kind of Chinese green vegetables.

Reviewer 4 Report
The current study provided a method for fast detecting pesticide residue by analyzing infrared microscopic images of brassica campestris leaves with computer vision methods.
How did you chose the number of samples in each set? (Figure 3)
Please explain why you obtain such difference in accuracy? (Table 2) For example: 21.21-92.36 -> 21 stands for fail and 92 for vary good performance.
Author Response
Response to Reviewer 4 Comments
We are very grateful for the your comments on our manuscript.
After carefully reading the comments, we revised the manuscript and invited a friend who is good at English to check language and grammar of the revised manuscript. All the modifications are marked in the text with yellow highlight. We hope that the correction will meet with approval.
The Point-by-point responses to the reviewers’ comments are as followings.
Point 1:How did you chose the number of samples in each set? (Figure 3)
Response 1: In line 412-414 of the revised manuscript, we introduced the idea of selecting the number of samples in each set.
Point 2:Please explain why you obtain such difference in accuracy? (Table 2) For example: 21.21-92.36 -> 21 stands for fail and 92 for vary good performance
Response 2: We are sorry for forgetting discussion the results of Naïve Bayesian method in the original manuscript. In line 500-504 of the revised manuscript, we added the discussions.

Round 2
Reviewer 2 Report
Revised satisfactorily.
Author Response
Dear Reviewer:
We are very grateful for the reviewers giving new comments on our revised manuscript (ID: sensors-2092269). After carefully reading the comments, we revised the manuscript and polished the language and grammar of the revised manuscript. All the modifications are marked in the text with green highlight. We hope that the correction will meet with approval.
The Point-by-point responses to the reviewers’ comments are as followings.
Waiting for your reply and have a nice day!
Best regards,
Yours sincerely,
Haoran Sun
Corresponding author:
Liguo Zhang
Email: zlgfyt@ecust.edu.cn
Point 1: English language and style are fine/minor spell check required
Response 1: Thank you for reminding it. We have revised the English expression in the text and highlighted it in green.

Reviewer 3 Report
I appreciate the changes introduced by the authors, including the revision of the writing. I have one major and two minor comments about the changes that they introduced.
The authors erroneously claim that 935 nm is equal to 1069 cm-1, while in reality it's 10695 cm-1, an order of magnitude error. The FTIR spectra of the vegetable and pesticides are irrelevant for this study, because the bands they show are fundamental vibrations in the mid-infrared, not overtones in the near-infrared. Therefore, this new information is erroneous and confusing. If the authors do not have NIR spectra in the correct range, they should not include any spectra and leave the identification of the bands as an open problem. In summary, Figure 3 is unacceptable in its present form and should be replaced or deleted.
Besides, I have two minor cosmetic comments about these changes. First of all, the authors replaced the correct term "absorbance" by "absorbency" several times. This should be reversed. Besides, they erased the scientific name of the plant, which I did not ask for. I suggest that they write "Shanghaiqing (Brassica rapa)" in lines 9 and 126 to clarify.
I appreciate the efforts of the authors and I'm ready to accept the article as soon as they implement these small changes.
Author Response
Dear Reviewer:
We are very grateful for the reviewers giving new comments on our revised manuscript (ID: sensors-2092269). After carefully reading the comments, we revised the manuscript and polished the language and grammar of the revised manuscript. All the modifications are marked in the text with green highlight. We hope that the correction will meet with approval.
The Point-by-point responses to the reviewers’ comments are as followings.
Waiting for your reply and have a nice day!
Best regards,
Yours sincerely,
Haoran Sun
Corresponding author:
Liguo Zhang
Email: zlgfyt@ecust.edu.cn
Point 1: Moderate English changes required.
Response 1: Thank you for reminding it. We have revised the English expression in the text and highlighted it in green.
Point 2: Are the methods adequately described?
Response 2: Thank you for pointing out this. We have added some supplementary notes to the method section in lines 199-205, 216-220, 235-242, 250-256 and 303-308.
Point 3: The authors erroneously claim that 935 nm is equal to 1069 cm-1, while in reality it's 10695 cm-1, an order of magnitude error. The FTIR spectra of the vegetable and pesticides are irrelevant for this study, because the bands they show are fundamental vibrations in the mid-infrared, not overtones in the near-infrared. Therefore, this new information is erroneous and confusing. If the authors do not have NIR spectra in the correct range, they should not include any spectra and leave the identification of the bands as an open problem. In summary, Figure 3 is unacceptable in its present form and should be replaced or deleted.
Response 3: Thank you for pointing out this mistake. We have changed Figure 3 to explain why we use 935 as the wavelength of light source and discussed it in line 199-205, and we have changed original reference 22 to a new reference that is correlated with present study.
Point 4:The authors replaced the correct term "absorbance" by "absorbency" several times. This should be reversed.
Response 4: Thank you for pointing out this. I have modified the errors in lines 154 and 155.
Point 5:Besides, they erased the scientific name of the plant, which I did not ask for. I suggest that they write "Shanghaiqing (Brassica rapa)" in lines 9 and 126 to clarify.
Response 5: Thanks for this reminding. I have changed it in lines 9and 132.
